# Increased Pre-Operative Lung Immune Prognostic Index Score Is a Prognostic Factor in Cases of Pathological T3 Renal Cell Carcinoma

**DOI:** 10.3390/curroncol32060335

**Published:** 2025-06-07

**Authors:** Hayato Hoshina, Toru Sugihara, Masayuki Kurokawa, Ei-ichiro Takaoka, Satoshi Ando, Haruki Kume, Tetsuya Fujimura

**Affiliations:** 1Department of Urology, Jichi Medical University Hospital, Shimotsuke 329-0498, Japan; hoshina.hayato@jichi.ac.jp (H.H.); kurokawa.masayuki@jichi.ac.jp (M.K.); takaoka.eiichiro@jichi.ac.jp (E.-i.T.); ando@jichi.ac.jp (S.A.); tfujimura@jichi.ac.jp (T.F.); 2Department of Urology, Graduate School of Medicine, The University of Tokyo, Tokyo 113-8657, Japan; kumeh-uro@h.u-tokyo.ac.jp

**Keywords:** lung immune prognostic index, prognostic factor, renal cell carcinoma, T3, immune biomarkers

## Abstract

We assessed the efficacy of the lung immune prognostic index (LIPI) in predicting the progression of pathological T3 renal cell carcinoma (RCC). The LIPI scores of patients with pathological T3 RCC were calculated in the pre- and post-operative phases. Patients were divided into zero-point, one-point, and two-point groups according to their LIPI score and into the upstage and non-upstage groups according to the pre- and post-operative increase in LIPI score. Overall survival (OS) was evaluated using Kaplan–Meier curves stratified by group. Univariate and multivariate analyses of OS were performed via Cox proportional hazard regression analysis. LIPI scores were calculated in 80 patients wherein blood sampling data were available. The upstage and non-upstage groups comprised eight and seventy-two patients, respectively. Kaplan–Meier curves showed a significant difference in the pre- to post-operative LIPI score upstage group. LIPI score change was a poor prognostic factor using univariate analysis (OS: hazard ratio (HR) = 4.10, 95% confidence interval (CI) = 1.07–15.61, *p* = 0.038) and multivariate analysis (OS: HR = 4.38, 95% CI = 1.13–16.89, *p* = 0.031). An increase in the LIPI score in the pre-operative phase was a poor prognostic factor for pathological T3 RCC.

## 1. Introduction

The advent of immune checkpoint inhibitors (ICIs) for cancer treatment represents a major shift in the chemotherapy era. They have resulted in significant changes in treatment regimens since the early days of chemotherapy. ICIs have been used for treating a wide range of cancers; numerous studies have been conducted to identify the factors that predict their efficacy in different cancer types. The lung immune prognostic index (LIPI), which was a score calculated by combining two blood-based parameters—derived neutrophil-to-lymphocyte ratio (dNLR) and lactate dehydrogenase (LDH)—has been studied as a predictor of patient responses to ICIs in non-small cell lung cancer cases [1,2,3,4,5]. Studies on LIPI have also been conducted for other cancer types, including renal cell carcinoma (RCC), to assess the predictive value of ICI in terms of response, overall survival (OS), and recurrence-free survival (RFS) [6,7,8,9].

Adjuvant treatment is now available for managing patients with RCC who are considered to be at a high risk of recurrence, as indicated by the KEYNOTE 564 trial [10,11,12,13,14,15]. Patients diagnosed with pathological T3 (pT3) have worse prognoses than those with T1 and T2, thus requiring more careful monitoring [16]. However, there are currently no tumor markers that reflect the disease status of RCC. The use of ICIs also exposes the patient to the risk of developing side effects. If serious side effects occur, there is a risk that the choice of treatment options at the time of relapse will be limited. We have therefore focused on LIPI score as a means of selecting patients for whom adjuvant treatment is more likely to be recommended and investigated whether changes in LIPI scores could be used to predict the progression of pT3 RCC.

In this study, we evaluated the pre- and post-operative LIPI scores in patients with pT3 RCC as well as changes in LIPI scores between the two periods. We also assessed whether LIPI scores and their degree of change could be associated with disease progression.

## 2. Methods

### 2.1. Patients

From January 2014 to July 2024, 545 patients with suspected renal cell carcinoma underwent surgical treatment at Jichi Medical University Hospital. In terms of surgical technique, partial or total resection, laparoscopic surgery, open surgery, and robotic surgery were selected according to tumor size and patient background. Of these, 85 patients diagnosed with pT3 were listed and 80 patients with appropriate blood samples were included (thirty-nine, thirty, six, and five patients who underwent open total nephrectomy, laparoscopic total nephrectomy, robot-assisted partial nephrectomy, and robot-assisted total nephrectomy, respectively). This study was a single-center, retrospective study.

### 2.2. Ethical Considerations

This study was approved by the Institutional Review Board of Jichi Medical University Hospital (approval no. A22-023). Written informed consent was obtained from all participants prior to their inclusion in the study.

### 2.3. Endpoints

The primary outcome was OS divided by the pre- and post-operative LIPI score and changes in LIPI scores between the two periods. OS was defined as the time from surgery until death.

### 2.4. Calculation of LIPI Score

LIPI score was calculated based on the patients’ blood test results. LIPI score was assessed by two parameters: dNLR and LDH. It was assigned on the basis of a dNLR (neutrophils/(leukocytes–neutrophils)) of >3 according to the cutoff used in the largest study published to date on ICI use in patients with cancer. The upper limit of LDH was 222 IU/L [1,17]. This LDH cutoff value was adopted by our institution, similar to what has been used in previous studies [6,18]. The LIPI score was assigned as follows: 2 points, dNLR > 3 and LDH > 222 U/L; 1 point, dNLR > 3 or LDH > 222 U/L; and 0 points, neither criterion met. Based on the LIPI score, the patients were stratified into three groups: the zero-point, one-point, and two-point groups.

### 2.5. Data Collection

The LIPI score calculated from each patient’s pre-operative blood sample (taken within one month pre-operatively) was defined as the pre-operative score, whereas the score calculated using the patient’s post-operative blood sample (taken within three months post-operatively) was defined as the post-operative score (Figure 1). The group was divided into two groups by LIPI score, one with zero points and the other with one or two points. Changes in LIPI scores between the pre- to post-operative scores were calculated and classified into two groups: upstage (0–1, 0–2, and 1–2) and non-upstage (other patterns).

### 2.6. Statistical Analysis

The OS between pre- and post-operative LIPI scores and between the upstage and non-upstage groups were assessed using the Kaplan–Meier method and compared via the log-rank test. The associations of OS with clinical parameters, including pathology results and LIPI score changes, were assessed using Cox’s proportional hazard regression analysis. The parameters included pre- and post-operative LIPI scores, as well as the changes thereof. The Fuhrman and WHO/International Society of Urological Pathology (ISUP) classifications were divided into grades 3/4 and others. Pre- and post-operative levels of serum C-reactive protein (CRP), which is considered a prognostic predictor of RCC, were also included with a cutoff of 0.5 [19,20,21]. Regarding univariate comparisons, categorical data were compared using the Chi-squared test; continuous data were compared using the Mann–Whitney U test. All statistical analyses were performed using JMP Pro, version 17 (SAS Institute, Cary, NC, USA). Statistical significance was set at *p* < 0.05.

## 3. Results

The patients’ background characteristics are presented in Table 1 (n = 80). None were treated with pre-operative chemotherapy; nine cases were treated with pembrolizumab as a post-operative adjuvant therapy.

The LIPI scores and cases at the pre- and post-operative phases are shown in Figure 2. Pre- and post-operative dNLR, LDH, and LIPI scores for all cases are shown in Appendix A. The pre-operative LIPI scores were 0 points in sixty cases, 1 point in sixteen cases, and 2 points in four cases. The post-operative LIPI scores were 0 points in sixty-seven cases, 1 point in eleven cases, and 2 points in two cases. The change in LIPI score from the pre- to post-operative periods corresponded to eight cases in the upstage group and 72 in the non-upstage group; the distribution of pre- and post-operative LIPI scores in each group is presented in Appendix A. The distribution of post-operative LIPI scores showed statistically significant differences, with a trend towards higher LIPI scores in the upstage group. In contrast, the non-upstage group had a higher proportion of cases of pre-operative LIPI scores of 2 points.

The OS was assessed using the Kaplan–Meier method; the pre- and post-operative LIPI scores as well as the change in LIPI score pre- to post-operative in two groups were compared using the log-rank test. The median OS for the pre-operative LIPI score was not reached in the 0-point group (95% confidence interval (CI) = 104—not reached (NR)) and the 1 point and 2-point groups (95% CI = 47—NR) (Figure 3A). The median OS was also NR in the 0 point group (95% CI = 104—NR) and the 1-point and 2-point groups (95% CI = 12—NR) in terms of post-operative LIPI score (Figure 3B). Considering the pre- to post-operative period, there were eight cases in the upstage group and seventy-two in the non-upstage group in terms of LIPI score change. The median OS was NR in the upstage group (95% CI = 104—NR) and non-upstage group (95% CI = 12—NR) (Figure 3C).

The log-rank test was used for evaluating the difference in OS between the two groups; no significant difference was found in pre-operative LIPI (zero points vs. one and two points) (*p* = 0.725), nor was a significant difference found in post-operative LIPI (zero points vs. one and two points) (*p* = 0.402). On the other hand, there was a statistically significant difference in the change in LIPI score from pre- to post-surgery (upstage group vs. non-upstage group) (*p* = 0.024).

Univariate and multivariate analyses of OS were performed using Cox proportional hazard regression (Table 2). Regarding OS, our univariate analysis identified sarcomatoid change and LIPI score change (pre- to post-operative) as poor prognostic factors. Our multivariate analysis of these factors identified both LIPI score change (pre- to post-operative) (hazard ratio (HR) = 4.38, 95% CI = 1.13–16.89, *p* = 0.031) and sarcomatoid change (HR = 8.71, 95% CI = 1.64–46.19, *p* = 0.010) as independent factors for poor prognosis.

## 4. Discussion

In this study, we investigated whether pre- and post-operative LIPI scores and the changes in these scores correlated with OS in patients with pT3 RCC. The change in LIPI score pre- to post-operative was found to highly correlate with OS. The results suggested that the change in LIPI scores could be used to predict the course of the disease, as well as a pathological diagnosis.

LIPI was first reported to represent a predictor of patient responses to ICIs, mainly in patients with non-small-cell lung cancer [1]. Its prognostic value in terms of ICIs has since been demonstrated in other cancers as well [6], including RCC [22,23] (Appendix A). Several studies have reported an association between LIPI score and both ICI and tyrosine kinase inhibitors in metastatic RCC cases [24,25,26]. Although the literature on LIPI is scattered and suggests an association with renal cancer progression and response to drug treatment, the number of reports examining changes associated with surgical treatment is limited. Therefore, this study examined how the LIPI score changes as a result of surgery.

The dNLR of LIPI has recently been studied in various ways. Although the neutrophil-to-leukocyte ratio (NLR) appears to represent a prognostic factor in patients with non-small cell lung carcinoma, it may be even more relevant than NLR because it includes monocytes and other granulocyte subpopulations [27,28]. High dNLR is associated with shorter survival times in patients with several tumor types—including melanoma, as well as pancreatic, bladder, and kidney cancer [17,29,30,31].

A number of studies have examined LIPI scores in blood test results at single points in time, such as before surgery or after ICI treatment. However, this study focused on how LIPI scores changed over the course of the RCC disease. This is because we hypothesized that the blood test results do not necessarily reflect the tumor environment only at the particular time point of sampling. Infections, inflammation, hematologic disorders, and oral steroid use, for instance, can also cause elevated white blood cell counts. We therefore suggest that performing multiple tests to make more accurate diagnoses is important. In this study, we evaluated the association with OS using only pre- and post-operative LIPI scores; however, no significant results were found. This study was also validated with respect to progression-free survival (PFS), without showing statistical significance in pre- and post-operative LIPI scores or in pre- and post-operative LIPI score changes. This difference in OS and PFS outcomes could be due to differences in treatment intervention and efficacy or owing to the site of recurrence.

This study only investigated LIPI scores in pT3 RCC. However, we recognize that other reports have evaluated this parameter in RCCs of the ≥pT3, N1-2, and M0 classifications as well—all of which are considered high risk [32]. In the KEYNOTE 564 trial, pT3 RCC was considered appropriate for treatment via adjuvant therapy, owing to its high risk of recurrence [10,11,12]. Several reports have shown that RCC with a pathological diagnosis of T3 has a worse prognosis compared with the T1 and T2 types [30,31,32]. Considering that T4 RCC cases are exceedingly rare, this study only included cases of RCC with pathological diagnoses of T3 following surgical treatment. In real clinical situations, how treatment is managed for patients with T3 renal cancer is very important. As not all patients would opt for adjuvant treatment, this study focuses on changes in the LIPI score to improve patient selection and treatment outcomes.

The lack of suitable tumor markers makes accurately monitoring RCC progression difficult. Several studies have reported that serum CRP level is associated with the progression and recurrence of RCC [33,34,35]. This present study also evaluated serum CRP levels in terms of OS using univariate analyses. Using univariate analyses, pre- and post-operative CRP was not an independent predictor of poor prognosis. In clinical practice, CRP level can be influenced by a number of factors; consequently, it is not considered to accurately reflect early-stage tumor effects. Therefore, monitoring disease progression using several prognostic markers over multiple time points is important.

This study has a few limitations that are worth noting. Selection bias might have occurred because the data were retrospectively obtained from a single center. In this study, the pathology of RCC was classified as clear cell type or other, and abnormal growth patterns such as rhabdoid changes and necrosis, which can impact prognosis, were not considered [36]. Both the Fuhrman and ISUP grading systems were used for nuclear grading. Although one of these systems is standardized, the system used at the time of diagnosis was also considered so as to reduce the burden on the pathologists [37].

## 5. Conclusions

In this study, patients with pT3 RCC wherein the LIPI scores were elevated from the pre-operative phase were found to generally have poor disease prognoses. There were no specific tumor markers for RCC, making accurately monitoring the microenvironment of this tumor type difficult. Monitoring changes in LIPI scores might thereby help manage the treatment course in patients with this deadly malignancy and might also be useful for selecting patients to whom more adjuvant therapy should be given.

## Figures and Tables

**Figure 1 curroncol-32-00335-f001:**
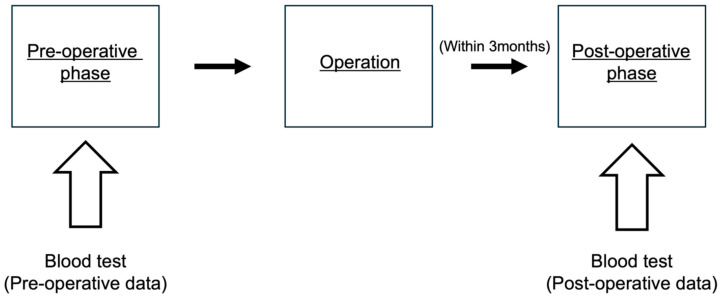
Timing of blood sampling used to calculate the patients’ LIPI scores. LIPI: lung immune prognostic index.

**Figure 2 curroncol-32-00335-f002:**
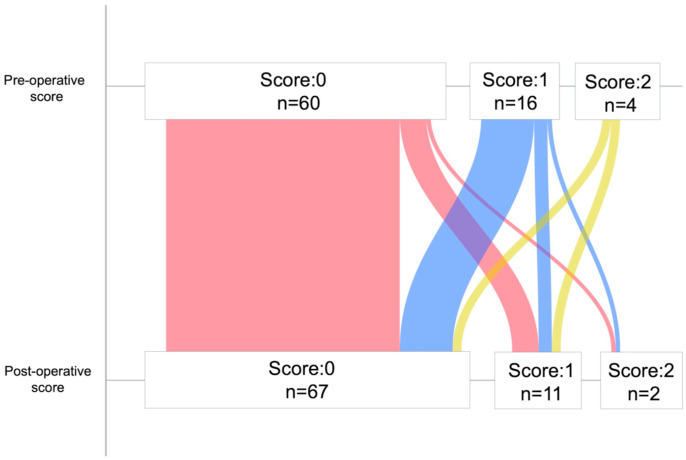
Number of scores and cases at the pre-operative phase and post-operative phase.

**Figure 3 curroncol-32-00335-f003:**
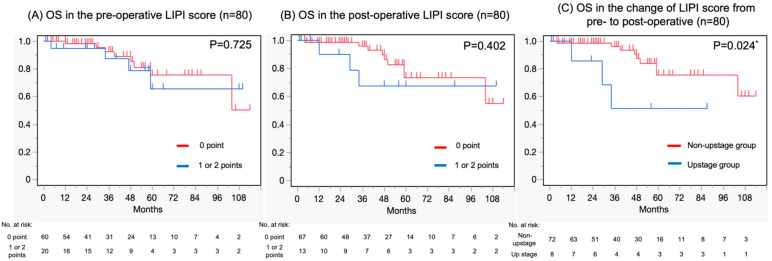
Kaplan–Meier curves depicting (**A**) OS in the pre-operative LIPI score, (**B**) OS in the post-operative LIPI score, and (**C**) OS in the change of LIPI score pre- to post-operative. OS, overall survival; LIPI, lung immune prognostic index. * Statistical significance.

**Table 1 curroncol-32-00335-t001:** Characteristics of the patient population (n = 80).

Parameter	Value (%) or Median (IQR)
Age, years	68 (60–75)
Sex:	
Male	59 (73.8)
Female	21 (26.2)
Surgical Technique:	
Open	39 (48.8)
Laparoscopic	30 (37.5)
RAPN	6 (7.5)
RARN	5 (6.2)
Tumor side, left	47 (58.8)
Pathological stage:	
pT3a	78 (97.5)
pT3b	2 (2.5)
Pathology:	
Clear cell	74 (92.5)
Non-clear cell	6 (7.5)
Fuhrman, or WHO/ISUP grade:	
1	7 (8.8)
2	43 (53.8)
3	19 (23.7)
4	8 (10)
Unclassified	3 (3.7)
Lymphatic vessel invasion	66 (82.5)
Sarcomatoid change	8 (10)
Pre-operative metastasis	9 (11.3)
CRP > 0.5	
Pre-operative	25 (31.3)
Post-operative	15 (18.8)

RAPN, robot-assisted partial nephrectomy; RARN, robot-assisted radical nephrectomy; ISUP, International Society of Urological Pathology; CRP, C-reactive protein; IQR, interquartile range.

**Table 2 curroncol-32-00335-t002:** Univariate and multivariate Cox proportional hazard regression analyses of OS in the study population (n = 80).

Parameter	Cutoff	Univariable		Multivariable	
		HR (95% CI)	*p*	HR (95% CI)	*p*
Sex	Male	2.78 (0.35–21.59)	0.328		
	Female	Reference			
Age	>70 years	1.51 (0.46–5.00)	0.492		
	≤70	Reference			
Fuhrman WHO/ISUP	Grades 3/4	1.15 (0.33–3.99)	0.814		
	Grade 1/2/unclassified	Reference			
Sarcomatoid change	Yes	7.81 (1.56–39.12)	0.012 *	8.71 (1.64–46.19)	0.010 *
	No	Reference		Reference	
Lymphatic vessel invasion	Yes	1.03 (0.22–4.70)	0.969		
	No	Reference			
Pre-operative LIPI	1 or 2 points	1.23 (0.37–4.13)	0.728		
	0 points	Reference			
Post-operative LIPI	1 or 2 points	1.73 (0.46–6.40)	0.411		
	0 points	Reference			
Pre-operative CRP	>0.5	1.29 (0.40–4.14)	0.662		
	≤0.5	Reference			
Post-operative CRP	>0.5	1.26 (0.34–4.71)	0.722		
LIPI score change (pre- to post-operative)	≤0.5 Upstage Non-upstage	Reference 4.10 (1.07–15.61) Reference	0.038 *	4.38 (1.13–16.89) Reference	0.031 *

OS, overall survival; CI, confidence interval; HR, hazard ratio; ISUP, International Society of Urological Pathology; CRP, C-reactive protein; LIPI, lung immune prognostic index. * Statistically significant.

## Data Availability

The dataset analyzed in this study is available from the corresponding author upon reasonable request.

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
