# Peer review of "Increased Pre-Operative Lung Immune Prognostic Index Score Is a Prognostic Factor in Cases of Pathological T3 Renal Cell Carcinoma"

_curroncol, 2025, doi:10.3390/curroncol32060335_

Round 1

Reviewer 1 Report

Comments and Suggestions for Authors Interesting research conducted in a very clear and schematic way.
The introduction, description of the methodology and presentation of the results are clear and understandable.
I have few tiny suggestions:
- add some details in LIPI score definition (lines 63 - 69) in order to clarify how to calculate the score
- add more details in the indices that carry the calculation of the LIPI score of the population presented
I suggest reporting the dNLP and the LDH of the 80 patients (pre and post-operative) clasterizing them later in the LIPI score.
- add a table in the discussion reporting the main study conducted on LIPI score and their results

Author Response

Comment 1: add some details in LIPI score definition (lines 63 - 69) in order to clarify how to calculate the score

Response 1: We have added the following sentence. Lines 65-66, 71-72.

Change in the text: LIPI score was assessed by two parameters: dNLR and LDH. LIPI score was assigned as follows: 2 points, dNLR > 3 and LDH > 222 U/L; 1 point, dNLR > 3 or LDH > 222 U/L; and 0 points, neither criterion met.

Comment 2: add more details in the indices that carry the calculation of the LIPI score of the population presented

Response 2: Thank you for your valuable comment. We have added more details on the calculation of the LIPI score to the Methods section. We now explicitly describe how the score was assigned: 2 points for both dNLR >3 and LDH >222, 1 point for either, and 0 points when neither criterion was met (lines 71-72) and add the explain of upstage and non-upstage criteria (lines 81-82).Change in the text: Changes in LIPI scores between the pre- to post- operative scores were calculated and classified into two groups: upstage (0–1, 0–2, 1–2) and non-upstage (other patterns).

Comment 3: I suggest reporting the dNLR and the LDH of the 80 patients (pre and post-operative) clasterizing them later in the LIPI score.

Response 3: Thank you for your comment. As you kindly suggested, the dNLR, LDH, the LIPI score and the results of determining the score change are included in Supplementary Table 1.

Change in text: Supplementary table 1.

Comment 4: - add a table in the discussion reporting the main study conducted on LIPI score and their results

Response 4: Thank you for your valuable suggestion. Some previous studies on LIPI, including other cancers, are listed in Table 3.

Change in text: Table 3

Reviewer 2 Report

Comments and Suggestions for Authors

The authors performed a retrospective, single-center study included 80 renal cell carcinoma (RCC) patients with pathological T3. The authors investigated whether changes in Lung Immune Prognostic Index (LIPI) scores could be used to predict the progression of pT3 RCC.

Some comments are listed below.

  1. The introduction should include more detail information of LIPI which combines two blood-based parameters: Derived Neutrophil-to-Lymphocyte Ratio (dNLR) and Lactate Dehydrogenase (LDH).
  2. The authors could consider to compare the non-upstage group (pre-operative LIPI score and post-operative LIPI score has significantly difference or not) as well as upstage group.
  3. In clinics, LIPI has been shown to correlate with progression-free survival (PFS). Do the authors also compare the PFS as Figure 3.
  4. iThenticate report: Percent match: 26%. The manuscript should revise to reduce percentage of matching to other articles.

Author Response

Comment 1: The introduction should include more detail information of LIPI which combines two blood-based parameters: Derived Neutrophil-to-Lymphocyte Ratio (dNLR) and Lactate Dehydrogenase (LDH).

Response 1: Thank you for your valuable suggestion. We have addedthe following sentence to lines 33-34 in the Introduction section and lines 71-72 in Methods section.

Change in text 1: The Lung Immune Prognostic Index (LIPI), which was a score calculated by combining two blood-based parameters: Derived neutrophil-to-lymphocyte ratio (dNLR) and lactate dehydrogenase (LDH), has been studied as a predictor of patient responses to ICIs in non-small cell lung cancer cases.

Comment 2: The authors could consider to compare the non-upstage group (pre-operative LIPI score and post-operative LIPI score has significantly difference or not) as well as upstage group.

Response 2: Thank you for your suggestion. The distribution of pre- and post-operative LIPI scores in the upstage and non-upstage groups has been presented in Table 2. We have also made an addition to the Result section. (lines 108-113)

Change in text 2: The change in LIPI score from the pre- to post- operative period corresponded to 8 cases in the upstage group and 72 in the non-upstage group; the distribution of pre- and post-operativeLIPI scores in each group is presented in Table 2. The distribution of post-operative LIPI scores showed statistically significant differences, with a trend towards higher LIPI scores in the upstage group. In contrast, the non-upstage group had a higher proportion of cases of a preoperative 2 score. Table 2

Comment 3: In clinics, LIPI has been shown to correlate with progression-free survival (PFS). Do the authors also compare the PFS as Figure 3.

Response 3: Thank you for your comment. This study did not find statistically significant differences in RFS. The median RFS for the pre-operative LIPI score was 53 months in the 0-points group (95% confidence interval [CI] = 42– not reached [NR]) and was not reached the 1-point and 2-points groups (95% CI = 6–NR) (Log-rank test: p=0.997). The median RFS for post-operative LIPI score was 48 months in the 0-points group (95% CI = 25–78) and the 1-point and 2-points groups was NR (95% CI = 4–NR) (Log-rank test: p=0.136).Regarding the pre- to post-operative period, the median RFS was NR in the upstage group (95% CI = 1–NR) and non-upstage group was 53 months (95% CI = 25–NR) (Log-rank test: p=0.366). We have added a note to the Discussion section regarding the RFS results. (Lines 170- 173)

Change in text 3: The study was also validated with respect to RFS, without showing statistical significance in pre- and post-operative LIPI scores or in pre- and post-operative LIPI score changes. This difference in OS and RFS outcomes could be due to differences in treatment intervention and efficacy or owing to the site of recurrence.

Comment 4: iThenticate report: Percent match: 26%. The manuscript should revise to reduce percentage of matching to other articles.

Response 4: Thank you for your comments. We have checked the iThenticate score and subtracted citations, references and preprint matches. If we excluded word-level matches with a match rate <1%, the result would be 12%. The result is even lower if we exclude matches other than textual content, such as author contributions andinformed consent. We therefore consider these results to be unproblematic.

Round 2

Reviewer 2 Report

Comments and Suggestions for Authors

Thanks for the revision. For the comment 3: The question is relating to progression free survival (PFS) not recurrence free survival (RFS).

Author Response

For the comment 3: The question is relating to progression free survival (PFS) not recurrence free survival (RFS).

Thank you for your comment.
I apologized for the misunderstanding regarding the terms 'Progression Free Survival' and 'Recurrence Free Survival'. As you rightly pointed out, PFS is the correct term.
It would be corrected.